# Two Sparse Matrices are Better than One: Sparsifying Neural Networks with Double Sparse Factorization

**Vladimír Boža**[1,2]  **Vladimír Macko**[1,3]

[1] Faculty of Mathematics, Physics and Informatics, Comenius University
[2] Powerful Medical   [3] GrizzlyTech
{boza, vladimir.macko}@fmph.uniba.sk

## Abstract

Neural networks are often challenging to work with due to their large size and complexity. To address this, various methods aim to reduce model size by sparsifying or decomposing weight matrices, such as magnitude pruning and low-rank or block-diagonal factorization. In this work, we present **Double Sparse Factorization (DSF)**, where we factorize each weight matrix into two sparse matrices. Although solving this problem exactly is computationally infeasible, we propose an efficient heuristic based on alternating minimization via ADMM that achieves state-of-the-art results, enabling unprecedented sparsification of neural networks. For instance, in a one-shot pruning setting, our method can reduce the size of the LLaMA2-13B model by 50% while maintaining better performance than the dense LLaMA2-7B model. We also compare favorably with Optimal Brain Compression, the state-of-the-art layer-wise pruning approach for convolutional neural networks. Furthermore, accuracy improvements of our method persist even after further model fine-tuning.

Code available at: `https://github.com/usamec/double_sparse`.

## 1 Introduction

Sparse neural networks have gained attention due to their potential to reduce computational costs and memory usage, making them more efficient for deployment on resource-constrained devices (LeCun et al., 1989; Han et al., 2015; Hoefler et al., 2021). By reducing the number of non-zero parameters, sparse networks can achieve accuracy similar to dense networks while requiring fewer operations. Reducing network size also decreases the number of weights that must be loaded into the processing unit from memory, which is crucial since memory bandwidth often becomes a bottleneck in neural network deployments, particularly during single-sample LLM inference (Xia et al., 2023).

In this work, we propose an improvement over a typical neural network sparsification. Instead of replacing each dense weight matrix with a sparse matrix, we replace each dense matrix with a product of two sparse matrices. We devise a heuristic algorithm for calculating sparse matrix factorization and achieve significant improvements over a wide range of models, including large language models and convolutional neural networks.

**Summary of contributions.** We propose a practical algorithm for factorizing a matrix into two sparse matrices called **Double sparse factorization (DSF)**. We extend it for the layer-wise pruning scenario where one wants to preserve layer behavior for a given set of calibration inputs. Our sparse factorization algorithm is a heuristic based on alternating minimization where each subproblem is solved using the ADMM algorithm for solving a sparse regression problem (Boža, 2024).

Our algorithm obtains superior results in the layer-wise pruning scenarios, where we fix the number of non-zero entries in each layer. We compare favorably to Optimal Brain Compression (Frantar & Alistarh, 2022) for pruning convolutional image models. We also produce state-of-the-art layer-wise pruning results for large language models, where the larger pruned model has better perplexity than the dense smaller model (as far as we know, this is the first time for the uniform layer-wise pruning).

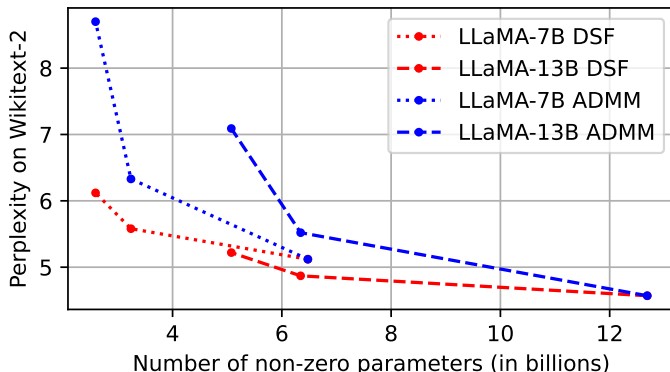

Figure 1: Comparison of LLaMA2 models pruned either using our DSF algorithm or using previously state-of-the-art ADMM pruning. We prune models using 0, 50, and 60% sparsities.

One could argue that our method requires storing one more pruning mask. We thus evaluate a scenario where one of the sparse factors mask (weights can be tuned, but nonzeros location is fixed) is randomly generated and fixed over the whole neural network. Our approach is better even in this scenario, which has almost the exact storage requirements as regular pruning.

Finally, we also show that our factorized pruning brings benefits even when sparsified models are further fine-tuned after pruning and achieve competitive results for pruning convolutional networks on CIFAR and ImageNet datasets.

## 2   RELATED WORK

**Neural network weight pruning and layer-wise one-shot pruning.** Post-training network pruning compresses the already training network by removing redundant weights (LeCun et al., 1989; Han et al., 2015; Blalock et al., 2020; Liu et al., 2018; Hoefler et al., 2021; Srinivas et al., 2022).

Some approaches focus on splitting the network into individual layer-wise problems, where one wants to preserve layer behavior over a small set of calibration inputs. Optimal Brain Compression (OBC) (Frantar & Alistarh, 2022) removes one weight at a time and optimally updates the remaining weights in the layer. However, this approach is not feasible for large language models due to high computational cost. SparseGPT (Frantar & Alistarh, 2023) uses various approximations and turns OBC into a more practical algorithm at the expense of higher approximation error. Wanda (Sun et al., 2023) proposes to skip the weight update and prune weights based on the product of absolute magnitude and input norm. Finally, Boža (2024) obtains state-of-the-art layer-wise pruning results using an ADMM-based algorithm, which uses gradual pruning combined with Wanda mask selection and ADMM (Boyd et al., 2011) weight update.

**Compression based on matrix factorization.** Instead of turning weight matrices into sparse matrices, one can replace them with a product of multiple smaller matrices. A typical example is a low-rank factorization (Li & Shi, 2018; Jaderberg et al., 2014) where one turns an $n \times m$ matrix into a product of $n \times k$ and $k \times m$ matrices, where $k << \min(n, m)$. More complicated examples include butterfly matrices (Dao et al., 2019) and Monarch matrices (Dao et al., 2022), where individual factors have some specific structure. Monarch matrices are the product of block-diagonal, permutation, and another block-diagonal matrix. The projection of a matrix into a set of monarch matrices is done by splitting the original matrix into blocks and then running a low-rank decomposition of each block. Another option is to decompose the matrix as a sum of low-rank and sparse matrix (Nikdan et al., 2024; Yu et al., 2017; Ke & Kanade, 2005; Wright et al., 2009). Distantly related to our work is sparse coding (Lee et al., 2006), which factorizes the matrix as a product of sparse and dense matrix to represent each data point (row of the matrix) as a linear combination of only a few basis vectors.

**Separable convolutions.** Convolutional layer can be naturally factorized into depthwise (applying filter per input channels) and pointwise convolution (mixing multiple channels). The idea was found initially in MobileNets (Howard, 2017; Sandler et al., 2018), but they placed nonlinearity between the depthwise and pointwise convolutions. However, some works successfully use separable convolutions without nonlinearity between them (Perešíni et al., 2021; Kriman et al., 2020).

**Sparse matrix factorization.** Factorization of the matrix into (sometimes more than two) sparse factors has already been studied. It was shown that this problem is NP-hard even when the sparsity pattern for factors is given (Le et al., 2021). Le Magoarou & Gribonval (2016) provides a heuristic based on the proximal gradient step called palm4msa, which is then used by Giffon et al. (2021) for compression of neural networks, but with very limited practical success. In the appendix, we compare the quality of our factorization algorithm with palm4msa. There were also works using sparse matrix factorization for parameter efficient fine-tuning (Chen et al., 2024).

## 3 PRELIMINARIES

In this work, we work with the post-training neural network sparsification scenario. We are given an already-trained network, and we will replace each weight matrix with a matrix that can be represented more efficiently, such as a sparse (Hoefler et al., 2021) or Monarch matrix (Dao et al., 2022). Usually, the replacement is done by solving the **projection** problem, where we are looking for a matrix closest (typically using the Frobenius norm) to the original one. For example, when the target matrix is sparse, solving the projection problem is just the magnitude pruning (Han et al., 2015).

In many cases, the sparsified network is often fine-tuned further. This can be prohibitive in some applications, especially involving large language models. We often resort to **one-shot** pruning in such cases. We capture relevant statistics for each layer and prune them during one forward pass. This is usually done by solving the **layer-wise pruning** problem (Frantar & Alistarh, 2022; 2023; Boža, 2024), where given calibration input $X$, original matrix $W$, one looks for sparse matrix $W_p$, such that the layer-wise error $||XW - XW_p||_2^2$ is minimized.

### 3.1 LAYER-WISE PRUNING VIA ADMM

Boža (2024) solves the layer-wise pruning problem by application of the alternating direction method of multipliers (Boyd et al., 2011) (ADMM). ADMM solves convex problems of the form: find minimum of $f(x) + g(y)$, subject to $Ax + By = C$, using iterations:

$$x^{k+1} = \arg\min_x f(x) + (\rho/2)||Ax + Bz^k - c + u^k||_2^2$$
$$z^{k+1} = \arg\min_z g(z) + (\rho/2)||Ax^{k+1} + Bz - c + u^k||_2^2$$
$$u^{k+1} = u^k + Ax^{k+1} + Bz^{k+1} - c$$

Note that when the pruning mask is fixed, the layer-wise pruning problem is a convex problem (we have one linear regression for each output with a different set of inputs).

This can be solved via ADMM as follows: Given $X$, $W$, and pruning mask $M$, we are looking for $W_p$ such that $(1 - M) \odot W_p = 0$ and layer-wise error is minimized. In ADMM formulation, $f(W)$ represents the layer-wise error, and $g(Z)$ would be an indicator function, which has a value of 0 when $Z$ has the correct mask and $\infty$ otherwise. Then we initialize $Z^0 = M \odot W$ and $U^0 = 0$ and apply following ADMM iterations ($\rho$ is penalty factor usually set to one, $U$ represents scaled dual variables):

$$\widehat{W}^{(k+1)} = (X^T X + \rho I)^{-1} (X^T XW + \rho(Z^{(k)} - U^{(k)})$$
$$Z^{(k+1)} = M \odot (\widehat{W}^{(k+1)} + U^{(k)}) \tag{1}$$
$$U^{(k+1)} = U^k + \widehat{W}^{(k+1)} - Z^{(k+1)}$$

We will take final $W_p = Z^{(m)}$ as the output.

Boža (2024) then applies the following improvements: Preconditioning is applied first to improve convergence. All input feature norms are normalized to one (and the matrix $W$ is multiplied by original input norms), which means that the diagonal of $X^T X$ contains only ones.

A pruning mask is found heuristically during the optimization process. During the first iterations, gradual magnitude pruning with cubic schedule (Zhu & Gupta, 2018) is applied (in the original paper, cubic prune is applied during the first 15 iterations out of 20; also it was found that dropping smallest values from $W^{k+1} + U^k$ is better than dropping smallest values from current valid solution $Z^k$).

## 4 DOUBLE SPARSE FACTORIZATION

In typical neural network pruning, we replace weight matrix $W$ with matrix $W_p$ which has at most $z$ nonzeros, i.e. $||W_p||_0 \le z$. Here, we propose to replace weight matrix $W$ with shape $n \times m$ with a product of two sparse matrices $AB$ such that they have at most $z$ nonzeros in total, i.e. $||A||_0 + ||B||_0 \le z$. We call this a **double sparse factorization**. Usually, we assume that $A$ is a matrix with shape $n \times n$, $B$ is a matrix with shape $n \times m$, and $n \le m$; if not, we transpose the matrix $W$.

During neural network inference, we multiply some input $X$ of shape $b \times n$ with matrix $W$. After our double sparse factorization which replaces $W$ with product $AB$, we will first multiply $X$ by $A$ and then by $B$, i.e. doing $(XA)B$. Note, that the total number of multiplications is $bz$, the same as in typical neural network pruning.

### 4.1 EXPRESSIVENESS AND EFFICIENCY OF DOUBLE SPARSE FACTORIZATION

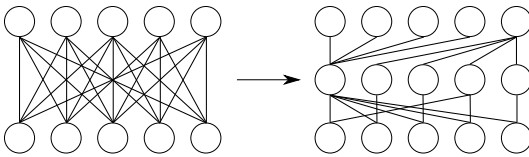

Figure 2: Graphical illustration of double sparse factorization. A dense layer is turned into two sparse layers. With enough weights in sparse matrices, most connections will be covered by a path through sparse matrices.

Many matrix factorizations mentioned previously in the literature can be (often trivially) rewritten to double sparse representation with the same number of non-zeros. For example, low-rank factorization commonly done via SVD (Li & Shi, 2018; Stewart, 1993) is already in the double sparse form. Monarch factorization (Dao et al., 2022), which is a product of block-diagonal, permutation, and block-diagonal matrices, can represented in double sparse form by fusing a permutation matrix with one of the block-diagonal matrices. Also, DSF can efficiently represent a matrix that consists of multiple disjoint low-rank submatrices.

The tricky case is an ordinary sparse matrix $W_p$. It can be represented in double sparse form as a product of identity and the original matrix: $IW_p$. However, this comes with the cost of additional non-zero entries for the identity matrix. Nevertheless, in the experiments, we will show that the double sparse representation represents the original dense matrix much better than an ordinary sparse matrix with the same number of non-zeros.

### 4.2 HEURISTIC ALGORITHM FOR DOUBLE SPARSE FACTORIZATION

First, we look into the **projection** problem. Given matrix $W$, we want to replace it with some compressed matrix $W_c$, where their difference $||W_c - W||_F$ is minimized.

In our case, $W_c$ is a product of two sparse matrices $A, B$. Thus, we are given matrix $W$ and are looking for matrices $A, B$ such that:

$$\text{minimize } ||AB - W||_F$$
$$\text{subject to } ||A||_0 + ||B||_0 \leq z$$

This problem is NP-hard even when the sparsity pattern for matrices $A$ and $B$ is given (Le et al., 2021) thus we solve our problem heuristically.

First, we decide how many nonzeros we allocate for each matrix, so our condition changes to $||A||_0 \leq z_a, ||B||_0 \leq z_b$. These allocations were determined manually in our experiments. In general, we found that it is beneficial to give one of the matrices approximately 1/3 of the nonzeros and 2/3 to the other one. Then, we continue with an alternating minimization algorithm. We fix the value of $A$ and find the best possible $B$, then fix the value of $B$ and try to find the best possible value of $A$. We repeat this process multiple times.

One inner step of our algorithm can formalized as: Given $W$, and $B$, find $A$ such that:

$$\text{minimize } ||AB - W||_F$$
$$\text{subject to } ||A||_0 \leq z_a$$

This problem is just an $L_0$ constrained linear regression. We solve it using an iterative ADMM solver Boža (2024) mentioned in preliminaries, which heuristically finds matrix mask and corresponding values. We also apply the following heuristic[1] improvements.

---

**Algorithm 1** Heuristical sparse matrix factorization for solving projection problem. Given matrix $W$, number of outer iterations $n$, number of inner iterations $m$ and number of nonzero elements $z_a, z_b$ we find $A, B$ such that $||A||_0 \leq z_a$, $||B||_0 \leq z_b$ and $AB$ is as close as possible to $W$.

---
Initialize $A^{(0)}, B^{(0)}$
$U_a^{(0)} = 0 \cdot A, U_b^{(0)} = 0 \cdot B$
**for** $k = 1..n$ **do**
    $\rho_0 = \min(1.0, k/(n-3))^3$
    $B^{(k)}, U_b^{(k)} \leftarrow$ solve $\arg\min ||AB - W||_F$, st. $||B||_0 \leq z_b$ via $m$ iterations of ADMM
             with starting point $B^{k-1}, U_b^{(k-1)}$ and starting $\rho_0$
    $A^{(k)}, U_a^{(k)} \leftarrow$ solve $\arg\min ||AB - W||_F$, st. $||A||_0 \leq z_a$ via $m$ iterations of ADMM
             with starting point $A^{k-1}, U_a^{(k-1)}$ and starting $\rho_0$
**end for**

---

**Warm starting the inner iterations.** To improve the convergence of ADMM iteration, we can warm-start it using the result from the previous step. To do that, we use not only the resulting sparse matrix but also all the dual variables $U$ from the ADMM algorithm. This allows us to decrease the number of inner iterations and speed up our algorithm.

**Annealing.** We found that our algorithm is often quickly stuck in some local optima. To prevent that, we propose a simple annealing scheme. The first step of ADMM for finding B is $\widehat{W}_b^{(1)} = (A^T A + \rho I)^{-1}(A^T W + \rho(B^{(0)} - U_b^{(0)}))$. Instead of using $\rho$ in the first iteration, we use smaller $\rho_0$ (we use default $\rho = 1$ in the remaining steps of ADMM). This gives lower weight to the previous solution and allows us to escape from local minima at the first steps of the optimization. We gradually increase $\rho_0$ from 0 to 1 throughout the optimization; we found that a simple cubic schedule works best. We also found that using more outer iterations ($n$) and fewer inner iterations ($m$) leads to better results.

**Initialization.** To run our algorithm, we must assign an appropriate starting values to matrices $A$ and $B$. We tested several choices (ablations are provided in the Appendix A.3), including random initialization and singular value decomposition, but we settled on initializing $A$ as an identity matrix and $B$ with magnitude pruning of the original input matrix.

---

[1] some people call this a dark magic

### 4.3 Application of Sparse Factorization to Layer-wise Pruning

Next, we look into **layer-wise pruning problem**. In our case of the sparse factorization, we are given calibration input $X$, original weight matrix $W$ and are looking for sparse $A, B$ such that the reconstruction error $||XW - XAB||_2^2$ is minimized.

We solve this problem by first running the weight projection algorithm from the previous section. However, for the pruning of LLMs, we found that it is better to project the weight matrix multiplied by input feature norms. This was previously done in Wanda pruning algorithm (Sun et al., 2023). We then scale one of the factors back. More formally we calculate matrix $W'$, such that $W'_{ij} = ||X_i||_2 \cdot W_{ij}$, then find $A'$ and $B$ such that error $||W' - A'B||_2^2$ is minimized and then compute $A_{ij} = \frac{1}{||X_i||_2} \odot A'_{ij}$. We do not do this rescaling for vision models.

We then proceed with the **finalization** step. We fix all sparsity masks and apply the ADMM algorithm for finding $B$ so that $||XW - XAB||_2^2$ is minimized. This is a straightforward modification of the ADMM algorithm.

However, finding $A$ is tricky and sometimes numerically unstable. In the inner iteration of ADMM, we need to find $A$ such that ($Z, U$ are other variables from ADMM optimization): $||XW - XAB||_2^2 + \rho/2||A - Z + U||_2^2$ is minimized. After taking gradients, we solve the equation: $X^T X A B B^T + \rho A = X^T X W B^T + \rho(Z - U)$. This is a special type of Sylvester equation Roth (1952); Jiang & Wei (2003), which can be solved using the eigendecomposition of $X^T X$ and $BB^T$. We provide a solution to this problem in the Appendix A.1. We found that optimizing $A$ is only helpful for compressing vision models; we do not use it when compressing large language models.

### 4.4 Computational considerations for DSF

The obvious drawback of DSF is having two sparse matrices compared to one. Here, we argue that doing computation with two sparse matrices needs resources comparable to doing computation with one sparse matrix with the same total number of nonzeros.

**Storage requirements.** Storing actual non-zero values has the same memory footprint for one sparse matrix and for DSF. The difference lies in storing nonzero positions (sparsity masks).

If we store sparsity masks as bit vectors, DSF would need to store two masks rather than one. This leads to a 2x increase in storage costs for square matrices but a smaller increase for rectangular matrices (for example, in Llama-2-7B, we have matrices with size 4096x11008, which would lead to a 37% increase in storage cost for masks). Remember that we also store nonzero values (usually 16-bit floats), and they take the majority of the storage costs. Overall, when measured on Llama-2-7B with 50% density, regular pruning would have a model size of 7.3GiB and DSF 7.7GiB. In the experiments, we show that DSF is still better even if we count the total model size.

The storage requirements will be identical if we store sparsity masks as nonzero positions (e.g., using a compressed sparse row format). However, this format is preferable only for higher sparsities since storing one position index usually takes 16 (or more) bits. There are also various other storage formats (e.g., delta coding), but they are unexplored in the context of neural network sparsity.

**Inference time.** One could think that doing two sparse multiplications would be much slower than doing just one because of the sparse matrix multiplication overhead. However, when looking at actual benchmarks, we find that in lower sparsity ranges (50-95%), doing two sparse multiplications is usually 10-20% slower than doing one sparse multiplication with the same number of nonzeros. For example, Xia et al. (2023) reports in their benchmark that multiplication with 60% sparsity takes 0.36s, and multiplication with 80% sparsity takes 0.21s, which would translate into 0.42s for DSF. Similar slowdowns around 10-20% can be found for GPU kernels done by Gale et al. (2020) and for CPU inference, which we tested in Appendix A.4.

We would like to point out that in the case of single sample LLM inference, the bottleneck is loading weights from memory to computational unit as reported in (Xia et al., 2023), and the current token activations can usually reside in the fast cache. Also, one goal of the pruning is to fit the model into available GPU memory, and in some cases, inference time can be sacrificed.

Table 1: Perplexity on Wikitext-2 for layer-wise pruning of large language models. Density refers to the total % of nonzero weights compared to the dense model.

| Density | Method | 1-7B | 2-7B | 2-13B | 2-70B |
|---------|--------|------|------|-------|-------|
| 100% | Dense | 5.68 | 5.12 | 4.57 | 3.12 |
| 50% | Wanda | 7.26 | 6.42 | 5.56 | 3.98 |
| | SparseGPT | 7.22 | 6.51 | 5.63 | 3.98 |
| | ADMM | 7.06 | 6.33 | 5.52 | 3.95 |
| | DSF | **6.12** | **5.58** | **4.87** | **3.44** |
| | DSF no fin. | 6.17 | 5.61 | 4.89 | 3.45 |
| | DSF one mask fix | 6.57 | 6.05 | 5.31 | 3.67 |
| 40% | Wanda | 10.66 | 9.71 | 7.75 | 4.98 |
| | SparseGPT | 10.51 | 9.58 | 7.80 | 4.98 |
| | ADMM | 9.22 | 8.70 | 7.09 | 4.81 |
| | DSF | **6.66** | **6.12** | **5.22** | **3.79** |
| | DSF no fin. | 6.76 | 6.29 | 5.32 | 3.81 |
| | DSF one mask fix | 7.82 | 7.47 | 6.21 | 4.27 |
| 30% | Wanda | 80.26 | 74.41 | 44.57 | 10.35 |
| | SparseGPT | 26.73 | 26.64 | 20.53 | 9.33 |
| | ADMM | 18.66 | 17.51 | 13.82 | 7.80 |
| | DSF | **8.33** | **8.01** | **6.43** | **4.56** |
| | DSF no fin. | 9.13 | 10.82 | 7.5 | 4.59 |
| | DSF one mask fix | 15.07 | 16.49 | 10.87 | 5.99 |

Table 2: Zero shot accuracies on zero-shot tasks when pruning Llama2-7B and Llama2-13B with 50% target density.

| Model | Method | BoolQ | RTE | HellaSwag | WinoGrande | ARC-e | ARC-c | OBQA | Mean |
|-------|--------|-------|-----|-----------|------------|-------|-------|------|------|
| | Dense | 77.71 | 62.82 | 57.19 | 69.22 | 76.35 | 43.43 | 31.40 | 59.73 |
| Llama2-7B | ADMM | 75.69 | 55.60 | 53.16 | **68.75** | 72.69 | 39.59 | **31.00** | 56.64 |
| | DSF | **75.72** | **57.04** | **55.22** | 67.32 | **75.51** | **42.83** | 30.80 | **57.78** |
| | Dense | 80.55 | 65.34 | 60.05 | 72.14 | 79.42 | 48.46 | 35.20 | 63.02 |
| Llama2-13B | ADMM | **81.35** | **64.98** | 56.70 | 72.53 | 76.52 | 43.00 | 33.20 | 61.18 |
| | DSF | 80.34 | 64.26 | **58.57** | **72.61** | **78.28** | **47.10** | **36.60** | **62.54** |

**Fine-tuning considerations.** Another concern is that during possible fine-tuning of the sparsified model, we need to store additional intermediate activations (in the middle of the double sparse factorization). This is true, but we found that with gradient checkpointing turned on and storing weights in compressed format (not as dense matrices), we can fine-tune on almost similarly sized sequences with DSF as when using regular pruning. We provide more details in the Appendix A.5.

## 5 EXPERIMENTS

We evaluate our proposed Double Sparse Factorization in multiple settings. First, we test it on layer-wise pruning of large language models. We compare our algorithms to ADMM pruning (Boža, 2024), which produces high-quality solutions in a reasonable time, even for large-scale models. Then we test it also on layer-wise pruning of vision models and compare it with Optimal Brain Compression (Frantar & Alistarh, 2022), state of the art layer-wise pruning algorithm. Finally, we also test whether image models compressed with DSF can be successfully fine-tuned.

We also include ablation studies of DSF in the Appendix A.3.

### 5.1 LAYER-WISE PRUNING OF LARGE LANGUAGE MODELS

**Setup.** We follow same setup as in Wanda (Sun et al., 2023) and ADMM pruning (Boža, 2024). We use 128 calibration samples from the C4 training dataset (Raffel et al., 2020) and prune layers

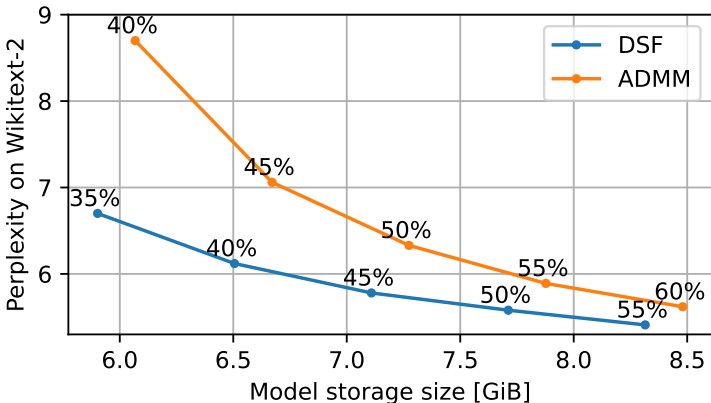

Figure 3: Comparison of total model size vs Wikitext2 perplexity for various target densities of Llama2-7B.

sequentially in order. We prune LLaMA (Touvron et al., 2023a) and LLaMA-2 (Touvron et al., 2023b) models. Similarly to previous works, we measure perplexity on held-out Wikitext (Merity et al., 2016). When factorizing square matrices (mainly in self-attention), we set the sparsity of one sparse factor to 16%. When factorizing rectangular matrices, the smaller factor will have 25% sparsity. The number of nonzeros in the other factor is just the target number of nonzeros minus the number of nonzeros in the first factor.

**Compared methods.** We compare our Double Sparse Factorization in three settings. The first one is the default one, solving the layer-wise pruning problem. Then, we disable the finalization step; thus, we only approximate the original dense matrix scaled by the input feature norms and solve the matrix projection problem. Finally, we fix one of the sparse masks to a random mask shared across all layers (but we run the finalization step). We compare our method with three layer-wise pruning algorithms: Wanda (Sun et al., 2023), which prunes weights with the smallest product of value and activation norm, ADMM pruning (Boža, 2024), which also updates weights during the pruning using alternating direction method of multipliers, and SparseGPT (Frantar & Alistarh, 2023).

**Results**. Results are summarized in Tab. 1 and Fig. 1. Our Double Sparse Factorization is superior to previous layer-wise pruning methods. To our knowledge, this is the first time when a uniformly layer-wise pruned network has better perplexity than its dense counterpart (compare 50% pruned LLaMA2-13B with perplexity 4.87 to dense LLaMA2-7B with perplexity 5.12). Even when we fix one mask (and thus make the total size of the network the same as in regular pruning), our factorization produces favorable results. We also notice that in the lower sparsities, the finalization step is not that important but becomes noticeably important at higher sparsities.

**Storage requirements for LLM vs its quality.** We also compare total storage size (nonzero values storage as 16-bit floats + binary mask) vs Wikitext2 perplexity. We measure various target densities for Llama2-7B. Results are shown in Fig 3. We can see that while DSF needs 0.4 GiB more storage for the same target density, it produces much better results, and this trend is more pronounced at lower densities.

**Results on zero-shot tasks.** We measure performance on seven zero-shot tasks (we use the same selection as the authors of ADMM pruning): BoolQ (Clark et al., 2019), RTE (Wang et al., 2018), HellaSWAG (Zellers et al., 2019), WinoGrande (Sakaguchi et al., 2021), ARC easy and challenge (Clark et al., 2018), and OpenbookQA (Mihaylov et al., 2018). Results are summarized in Tab. 2. On average, we obtain better performance than previous pruning methods.

**Pruning speed.** We can prune the 7B models in apx. 30 minutes on one Nvidia 4090 GPU (this includes both forward pass and sparse factorization times). Note that reported total running times for ADMM pruning and SparseGPT are around 10-15 minutes (Boža, 2024). We also study the effect of a number of iterations on pruning speed and solutions quality in the Appendix A.6.

Table 3: Comparision of our Double Sparse pruning vs. Optimal Brain Compression on Resnet50 using Imagenet dataset.

| Number of nonzeros | FLOP reduction | Method | Test accuracy [%] |
|---|---|---|---|
| 25.5M | - | Dense | 76.13 |
| 16.8M | 2x | OBC | 75.65 |
| | | DSF | **75.78** |
| 12.3M | 3x | OBC | 75.01 |
| | | DSF | **75.56** |
| 10.2M | 4x | OBC | 74.05 |
| | | DSF | **74.95** |

Table 4: Test accuracy on CIFAR-10 using Resnet-20 with varying width. Density refers to the total % of nonzero weights compared to the dense model. FT refers to fine-tuning.

| Density | Method | Resnet-20-16 | Resnet-20-32 |
|---|---|---|---|
| 100% | Dense | $92.2 \pm 0.2$ | $94.0 \pm 0.1$ |
| 20% | Magnitude w/o FT | $70.6 \pm 0.5$ | $86.2 \pm 0.3$ |
| | Double sparse w/o FT | $80.0 \pm 0.9$ | $91.6 \pm 0.1$ |
| | Magnitude w/ FT | $91.2 \pm 0.2$ | $93.5 \pm 0.1$ |
| | Double sparse w/ FT | $\mathbf{91.5 \pm 0.1}$ | $\mathbf{93.6 \pm 0.1}$ |
| 10% | Magnitude w/o FT | $29.0 \pm 2.9$ | $51.1 \pm 5.2$ |
| | Double sparse w/o FT | $48.9 \pm 2.9$ | $84.1 \pm 0.5$ |
| | Magnitude w/ FT | $89.3 \pm 0.3$ | $92.6 \pm 0.1$ |
| | Double sparse w/ FT | $\mathbf{89.8 \pm 0.2}$ | $\mathbf{93.0 \pm 0.2}$ |

**Further fine-tuning.** We also tested the possibility of further fine-tuning of pruned LLMs. DSF retains its advantage even after fine-tuning. Setup and results are summarized in Appendix A.7.

## 5.2 COMPARISON WITH OPTIMAL BRAIN COMPRESSION

Optimal Brain Compression (Frantar & Alistarh, 2022) is a post-training layer-wise pruning algorithm, which prunes each network layer by removing one connection at a time and optimally updating the remaining weights. Compared to the ADMM update algorithm mentioned in the previous section, it is much more accurate, but at the expense of much longer running time, unsuitable for large language models. However, OBC is still usable for moderately sized vision neural networks like ResNet50 (He et al., 2016).

In this experiment, we evaluate the effectiveness of our Double Sparse Factorization of ResNet50 on Imagenet (Russakovsky et al., 2015) dataset. We first run the OBC pipeline to determine layer-wise pruning ratios. Using the same calibration dataset as OBC, we then factorize every convolutional layer into two sparse matrices with the same number of nonzero weights as the OBC solution. We treat convolutions as linear layers, where input is processed via the im2col procedure. The sparsity of the smaller factor is set to $max(0.16, s/2)$ where $s$ is sparsity from OBC. The bigger factor will get the remaining nonzeros (so the total nonzeros of sparse factors match the number of nonzeros used by OBC). Results are summarized in Tab. 3. We see that our solution is superior to the solution found by OBC for every sparsity setting, and the gap grows wider with larger sparsities.

## 5.3 FINE-TUNING OF IMAGE MODELS PRUNED WITH DOUBLE SPARSE FACTORIZATION

Finally, we test whether the Double Sparse Factorization accuracy advantage remains after fine-tuning a whole model. In this experiment, we only focus on the original matrix projection and do not perform any input-dependent finalization. We test the pruning of Resnet-20 (He et al., 2016) with varying starting widths (16 and 32) on the CIFAR-10 (Krizhevsky et al., 2009) dataset. We also test pruning Resnet-50 on Imagenet dataset (Russakovsky et al., 2015). In all experiments, we use the same sparsity in all layers. For CIFAR-10 experiments, we first train the dense network using the

Table 5: Test accuracy on Imagenet using Resnet-50. Density refers to the total % of nonzero weights compared to the dense model. FT refers to fine-tuning.

| Density | Method | Test accuracy [%] |
|---|---|---|
| 100% | Dense | 76.13 |
| 20% | Magnitude w/o FT | 54.43 |
| | Double sparse w/o FT | 71.85 |
| | Magnitude w/ FT | 75.43 |
| | Cyclical pruning Srinivas et al. (2022) | 75.3 |
| | Double sparse w/ FT | **75.78** |
| 10% | Magnitude w/o FT | 9.87 |
| | Double sparse w/o FT | 55.76 |
| | Magnitude w/ FT | 73.32 |
| | Cyclical pruning Srinivas et al. (2022) | 73.3 |
| | Double sparse w/ FT | **74.50** |

procedure from Liu et al. (2022). We train for 160 epochs using SGD with a starting learning rate of 0.1 and 0.9 momentum. We decay the learning rate by 10 on epochs 80 and 120. We then prune each layer (except the first and last one) to 10 or 20% of nonzeros using either magnitude pruning or our double sparse factorization method (on the weight projection problem). Then, we fine-tune the model for 50 epochs, starting with a learning rate of 0.1 and linearly decay the learning rate to zero (this was inspired by (Zimmer et al., 2021)). We run each setting 5 times using different seed. Results are shown in Tab 4.

For the Imagenet experiment, we start with the pre-trained Resnet-50 from Torchvision (maintainers & contributors, 2016). We then uniformly sparsify every layer except the first and last one and fine-tune for 20 epochs using SGD, with a linear learning rate decay from 0.01 to zero and momentum of 0.9. We also compare with results reported by Srinivas et al. (2022), which prunes and fine-tunes the neural network in multiple cycles with resets (Cyclical pruning). Results are shown in Tab 5.

In all cases, starting test accuracy is higher for double sparse pruning and stays better when fine-tuned. This is especially evident at higher sparsities.

## 6 CONCLUSIONS AND FUTURE WORK

In this work, we introduced Double Sparse Factorization (DSF), an approach to decompose weight matrices into two sparse matrices, enabling more efficient neural networks. By applying DSF, we significantly improved layer-wise pruning for both large language models (LLMs) and convolutional neural networks (CNNs). The method effectively reduced the number of parameters without sacrificing model accuracy, achieving state-of-the-art results compared to traditional pruning techniques. Furthermore, our approach kept its performance gains even after further fine-tuning. Our work is also one of the first to show that a sparse neural network can achieve more gains by employing a more complicated technique than just removing weights. One drawback of our solution is that individual layer-wise sparsities need to be determined manually beforehand (compared to magnitude pruning, which can work globally and determine sparsity in each layer automatically). Also, it is unclear how to integrate DSF with gradual pruning with fine-tuning the whole network between pruning steps. We leave these enhancements for future work.

ACKNOWLEDGMENTS

This research was supported by grants 1/0140/25, and 1/0538/22 from Slovak research grant agency VEGA. Part of the research results was obtained using the computational resources procured in the national project National competence centre for high performance computing (project code: 311070AKF2) funded by European Regional Development Fund, EU Structural Funds Informatization of society, Operational Program Integrated Infrastructure. We would also like to thank Elvir Crnčević for his GPU kernel implementation for DSF (`https://github.com/elvircrn/double_sparse_kernel`).

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

## A  APPENDIX

### A.1  SOLVING FOR A IN THE LAYER-WISE RECONSTRUCTION PROBLEM

Recall that we want to find sparse $A$ such that $||XW - XAB||_2^2$ is minimized (where $X$ is calibration input, $W$ is the original weight matrix, and $B$ is the other sparse factor).

In the inner iteration of the ADMM, we need to find $A$ such that ($Z, U$ are other variables from ADMM optimization): $||XW - XAB||_2^2 + \rho/2||A - Z + U||_2^2$ is minimized.

After taking gradients, we solve the equation:

$$X^T X A B B^T + \rho A = X^T X W B^T + \rho(Z - U)$$

We solve this equation using eigendecomposition and one simple trick. We use following eigendecompositions: $X^T X = Q D Q^T, B B^T = R E R^T$ (where $D, E$ are diagonal matrices and $Q, R$ are orthonormal).

We then multiply the equation by $Q^T$ from left and $R$ from right and get:

$$D Q^T A R E + \rho Q^T A R = Q^T (X^T X W B^T + \rho(Z - U))R$$

We will now use that $D, E$ are diagonal and create an outer product of their diagonals: $F = \text{Tr}(D) \otimes \text{Tr}(E)$. Now, we can use Hadamard product to get:

$$F \odot Q^T A R + \rho Q^T A R = Q^T (X^T X W B^T + \rho(Z - U))R$$

And with slight abuse of notation (where $F + \rho$ means adding $\rho$ to every element of $F$) we get:

$$Q^T A R = Q^T (X^T X W B^T + \rho(Z - U))R \oslash (F + \rho)$$

And thus:

$$A = Q(Q^T (X^T X W B^T + \rho(Z - U))R \oslash (\text{Tr}(D) \otimes \text{Tr}(E) + \rho))R^T$$

### A.2  COMPARISON WITH OTHER MATRIX APPROXIMATION METHODS

We evaluate multiple methods for the weight projection problem. We use weight matrices from Llama-7B and Resnet-50. We truncate them to square matrices with sizes 64, 256, 1024, or 4096 (to accommodate Monarch factorization without problems). We evaluate our Double Sparse Factorization, palm4msa from Faust library (Le Magoarou & Gribonval, 2016), which also factories matrix into two sparse matrices, magnitude pruning, which keeps values with the largest magnitude, singular value decomposition, which factorizes matrix into two low-rank matrices, and Monarch decomposition (Dao et al., 2022), which factorizes matrix into block-diagonal, permutation and block-diagonal matrix. In all cases, we aim for 4x compression, i.e., each method can produce matrices that contain at most 25% of non-zeros in total compared to the original matrix.

Results are summarized in Fig. 4. We see that our DSF consistently outperforms other methods. Interestingly, palm4mse is not better for small matrix sizes than magnitude pruning. Also, Monarch decomposition seems to be worse than ordinary SVD.

### A.3  ABLATION OF DSF SETTINGS

We investigate some variations of DSF settings in Fig. 5. As in the experiments section, we target to have 25% of nonzeros compared to the original matrices. Running shorter iterations, especially our cubic first iteration weight schedule, benefits the final result. We also provide ablations for varying size of one factor (Fig. 6), and also for various initializations (Fig 7).

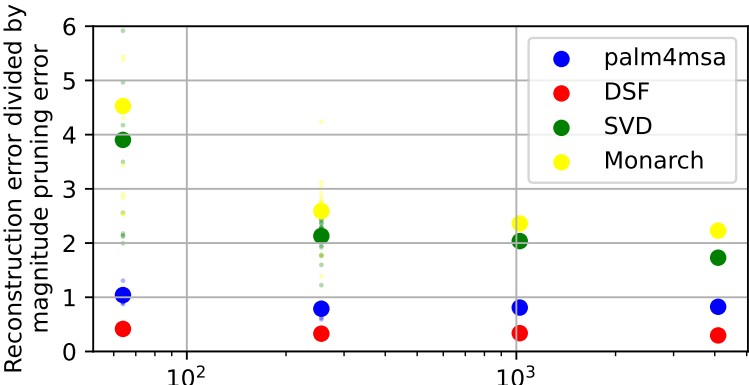

Figure 4: Reconstruction error of various compression methods on various matrix sizes for weight projection problem. We compress each matrix to 25% of the original size. We normalize error by error of magnitude pruning. The mean is denoted by a large dot and individual results with smaller dots.

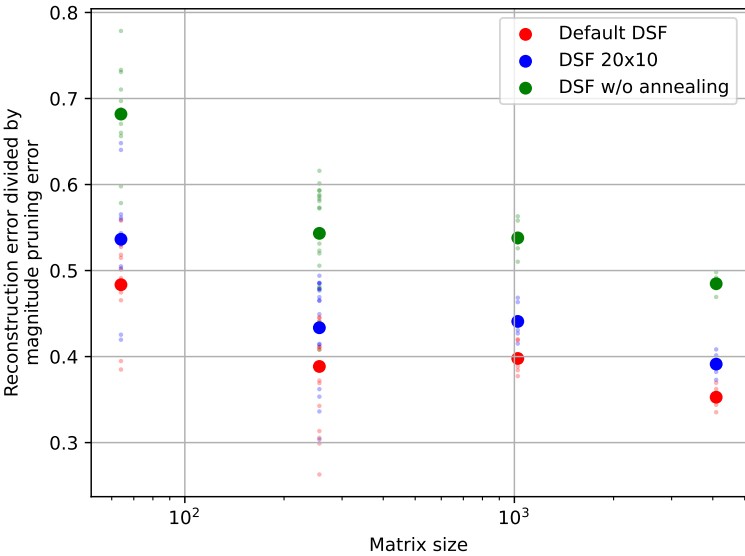

Figure 5: Reconstruction error of various settings of DSF. Default DSF used 40 outer and 5 inner iterations. DSF 20x10 refers to DSF with 20 outer and 10 inner iterations. DSF w/o annealing refers to DSF where we set first $\rho_0 = 1$.

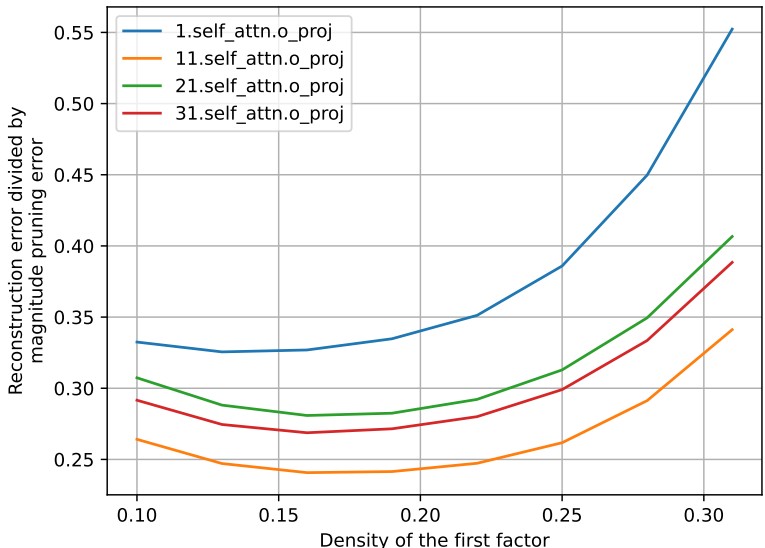

Figure 6: Reconstruction error for various densities of the first factor. We prune output projection from the attention layer in Llama-1 (matrix size 4096x4096) with a target total density of 40%. We see that the optimal density of the first factor is slightly less than half of the target total density.

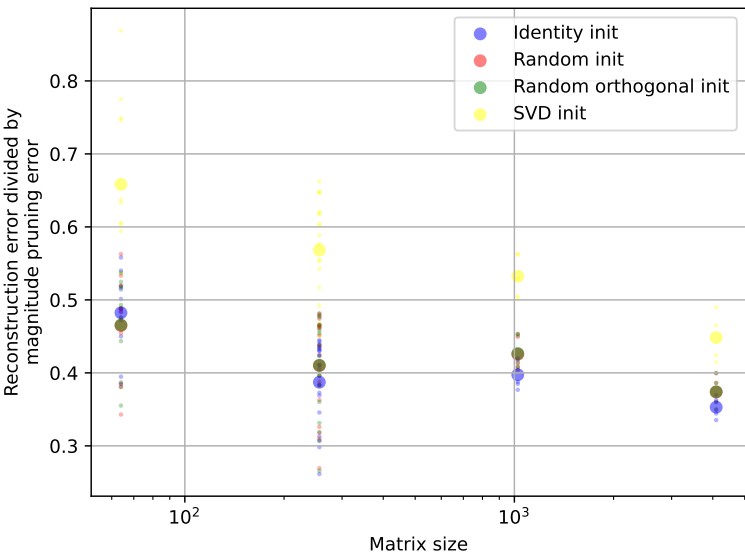

Figure 7: Reconstruction error for various initialization schemes of the first factor in DSF. We see that our choice of using identity init outperforms other common choices on all except the smallest matrices.

Table 6: Comparision of single core CPU runtime for simple sparsity and DSF. Each time, we compare the runtime of 48 layers with simple sparsity or DSF with an equal number of nonzeros (i.e., running 96 layers with half the density). We run the experiment on a single core using the DeepSparse engine.

| Layer width | Batch size | Dense time [ms] | Density | Simple sparsity time [ms] | DSF time [ms] |
|---|---|---|---|---|---|
| 1024 | 64 | 49.5 | 0.5 | 30.5 | 36.3 |
| | | | 0.25 | 18.9 | 21.3 |
| | | | 0.125 | 10.11 | 11.43 |
| | 256 | 191.5 | 0.5 | 108.1 | 129.2 |
| | | | 0.25 | 70.0 | 84.0 |
| | | | 0.125 | 43.2 | 47.2 |
| 4096 | 64 | 872 | 0.5 | 470 | 533 |
| | | | 0.25 | 282 | 310 |
| | | | 0.125 | 166 | 184 |
| | 256 | 3120 | 0.5 | 1734 | 1965 |
| | | | 0.25 | 1050 | 1132 |
| | | | 0.125 | 614 | 650 |

Table 7: Comparision of multi-core CPU runtime for simple sparsity and DSF. Each time, we compare the runtime of 48 layers with simple sparsity or DSF with an equal number of nonzeros (i.e., running 96 layers with half the density). We run the experiment on 8 cores using the DeepSparse engine. We throttled the frequency of the CPU to 3200 MHz to get consistent results unaffected by thermal throttling.

| Layer width | Batch size | Dense time [ms] | Density | Simple sparsity time [ms] | DSF time [ms] |
|---|---|---|---|---|---|
| 1024 | 64 | 11.3 | 0.5 | 5.43 | 6.74 |
| | | | 0.25 | 3.34 | 4.30 |
| | | | 0.125 | 2.10 | 2.70 |
| | 256 | 36.25 | 0.5 | 19.30 | 23.66 |
| | | | 0.25 | 11.87 | 15.66 |
| | | | 0.125 | 7.75 | 9.29 |
| 4096 | 64 | 162 | 0.5 | 101 | 122 |
| | | | 0.25 | 62.3 | 74.5 |
| | | | 0.125 | 37.7 | 44.93 |
| | 256 | 551 | 0.5 | 297 | 355 |
| | | | 0.25 | 177 | 221 |
| | | | 0.125 | 111 | 136 |

## A.4 SPARSE MATRIX EFFICIENCY ON CPU

We use the following benchmark setup: We create a chain of 48 or 96 matrices of size 1024x1024 or 4096x4096 (the goal here is that all matrices do not fit into the cache and must be reloaded during inference).

We then benchmark dense and sparse matrix multiplication over varying batch sizes and sparsity. We run on Intel(R) Core(TM) i7-11800H and use DeepSparse inference engine (NeuralMagic, 2021). We try to simulate DSF vs single sparsity comparison (e.g. compare 48 layers with 50% density vs 96 layers with 25% density). Results are summarized in Tab. 6 and 7. We see that DSF has mostly a runtime 10-20% longer than ordinary sparse matrix multiplication but still better than dense multiplication.

Table 8: Maximum sequence length for various fine-tuning setups for Llama2-7B using A100 with 40GB of memory and batch size 16. We always have gradient checkpointing turned on.

| Setup | Maximum sequence length |
|---|---|
| Dense (base Llama2-7B) | 1900 |
| 50% sparsity | 2600 |
| DSF with 50% total density | 2400 |

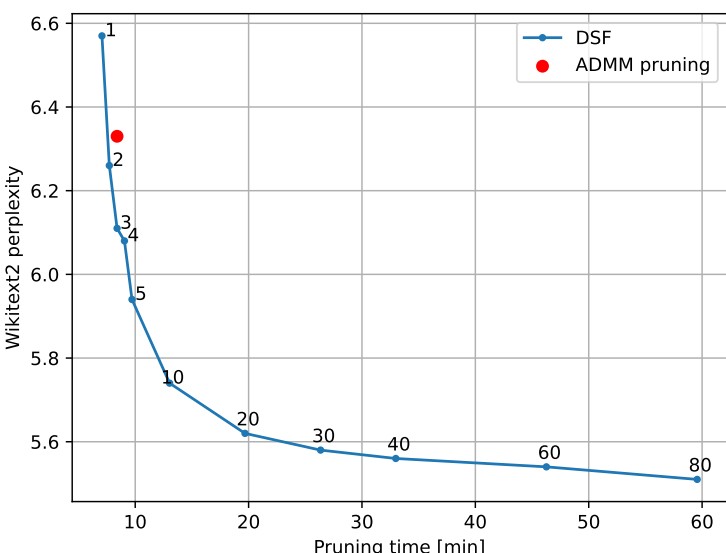

Figure 8: Comparison of total pruning time for DSF vs Wikitext2 perplexity for various number of pruning iterations. We also provide ADMM pruning for reference.

### A.5 EFFECTS OF DSF ON FINETUNING

DSF makes the network deeper, and during fine-tuning, one needs to store more intermediate activations. Moreover, if DSF results are stored as dense matrices, network size also increases.

We propose a simple storage solution for sparse matrices in Pytorch. We store nonzero values as parameters and packed masks as an auxiliary parameter. During the forward pass, we unpack everything into the dense matrix and then process the input. This incurs small time overhead (which gets smaller the bigger batch becomes). A custom kernel would obviously be a better solution, but writing custom GPU kernels is not the focus of this paper. This solution also only works if gradient checkpointing is enabled because otherwise, all of the unpacked dense matrices would be stored during the forward pass at once.

Here, we search for the maximum sequence length we can fine-tune. We finetune Llama2-7B using batch size 16 on A100 GPU with 40GB of memory. We use SGD optimizer without momentum. We turn on gradient checkpointing and test dense (vanilla model), sparse, and DSF representations. As reported in Tab. 8, when using DSF, the maximum fine-tunable sequence length drops, but it is still higher than when using dense representation.

### A.6 COMPARISON OF FACTORIZATION TIME VS MODEL QUALITY

Varying the number of DSF iterations can lead to smaller or larger pruning times. Here, we study the effect of the number of pruning iterations (and thus pruning time) on solution quality. We use

Table 9: Results after further LLM fine-tuning.

| | w/o finetuning | | w/ finetuning | |
|---|---|---|---|---|
| Starting point | Perplexity | Zeroshot | Perplexity | Zeroshot |
| Llama2-7B Dense | 5.12 | 59.71 | - | - |
| Llama2-7B ADMM 50% | 6.33 | 56.64 | 5.61 | 58.00 |
| Llama2-7B DSF 45% | 5.78 | 57.03 | 5.35 | 59.00 |

Llama2-7B with 50% target density and vary the number of factorization iterations from 1 to 80 (40 being the default). Results are summarized in Fig. 8

## A.7 FURTHER LLM FINETUNING

We also tested a limited full model fine-tuning of pruned LLMs. We finetune for two days on 4 A100 GPUs. We use a setup similar to (Malinovskii et al., 2024) and use knowledge distillation loss (results from Muralidharan et al. (2024) also suggest that distillation loss is just enough) and 1B sample of the Redpajama dataset (Weber et al., 2024) as the calibration dataset. We used batch with 1M tokens and AdamW optimizer.

We compare ADMM pruned Llama2-7B with 50% density and DSF pruned model with 45% density to compare the models with approximately the same storage sizes. Results are summarized in Tab. 9. DSF maintains its advantage even after fine-tuning.

