# OpenReview forum: "Two Sparse Matrices are Better than One: Sparsifying Neural Networks with Double Sparse Factorization"
_ICLR.cc/2025/Conference — ICLR 2025 Poster_

### Official Review · Reviewer_aP2Z · 2024-10-25

**Soundness:** 2
**Presentation:** 2
**Contribution:** 2
**Rating:** 5
**Confidence:** 4

**Summary:**

The paper proposes Double Sparse Factorization, a method that, instead of pruning the original weight matrix, factorizes it into the product of two matrices (similar to e.g. low-rank decomposition), which together satisfy the same sparsity constraint. To solve this problem, they use the ADMM method. The paper claims to improve upon existing pruning and layer-wise pruning approaches, and they back their claims with experiments on state-of-the-art language models and medium-sized vision models. In addition, they show that the superiority of their methods seems to prevail after retraining the pruned models.

**Strengths:**

The idea is interesting, to the best of my knowledge relatively novel, and the experiments are quite convincing. Most of the paper is fairly easy to follow and the reader is not left with many questions. I appreciate that the authors provide results before and after retraining the pruned models, as this is often not done in other papers. The proposed method is interesting, however there are open questions that I will discuss below.

**Weaknesses:**

I have several concerns regarding the soundness, clarity, and contribution of this work, which I detail below. I hope these remarks are helpful for improving the paper and am open to discussing my evaluation.

### Clarity
While I think that the idea proposed in this paper might be promising, I sometimes had a hard time following the paper. I think the structure as well as details could be improved.
- Section 3.1 would greatly benefit from a more detailed explanation of the ADMM method. How are Z and U initialized? I understand that it is not your job to explain ADMM in detail, but I think that nevertheless the paper would greatly benefit from more detailed remarks - at least in the appendix. Since this method is not standard (at least in the pruning literature and to my knowledge), I think it would be helpful to make this more clear.
- In two sentences (Line 149, 150) you basically explain how you find the sparsity mask. Why do you precondition? How exactly is the cubic schedule (I presume Zhu & Gupta?) implemented, over how many iterations, with which interval between the increases? I am trying my best to infer this from somewhere, but it is nowhere to be found? Either I am missing something or the paper is lacking a crucial part, namely how the sparsity mask is found.
- In Line 258, you state that you are using the Wanda saliency map, I think it would be good to give the mathematical formulation to that, especially how you "scale one of the factors back".


### Soundness
- Lines 37-39: If you replace the dense weight matrix with a product of two sparse matrices, will your model not be much slower at inference than when replacing with just a sparse matrix? For Low-rank decomposition, you at least get two linear layers which are much smaller dense matrices, but in your case, you basically have two sparse matrices. While you argue in Line 162 that the total number of multiplications is equal, this is far from realizable on the existing hardware. In practice, you incur a non-trivial overhead. I would like to hear the authors' thoughts on this.
- Line 50: "our method is the first layer-wise pruning method in which the larger pruned model is better than the dense smaller model" - Are you sure this is true? I feel like already the original SparseGPT paper gets fairly close and there have been a variety of improvements since then, e.g. using non-uniform layer-wise sparsity. Maybe this claim can or should be made more precise.


### Experimental Validation
- Missing ablations: The paper is fixing a lot of hyperparameters and making claims without ablations. That includes e.g. the selection of sparsity distribution between the matrices (Line 209) or the initialization for A and B (Lines 248-250), among others. Such ablations should be added to justify the choice of parameters.
- Table 1: Why are you not comparing to SparseGPT, am I missing something? In my experience, SparseGPT is a very strong baseline. Also, why do you omit Wanda for 30% density? Is Wanda using a "finalization" step as well, i.e., are you reconstructing the remaining weights after pruning? You get that more or less for free if you pass the calibration data through anyway.
- Section 5.4: I find the choice of hyperparameters for the retraining/fine-tuning quite arbitrary. You use a stepped schedule for most of the pretraining, then use a stepped learning rate schedule for retraining as wellf 70 total) epochs. [1] shows that if you properly choose the initial learning rate of a linear schedule, you can recover the accuracy drop of magnitude pruning in very few iterations. I am not sure if these results would withstand scrutiny. It would be good to use best practices here, i.e., for the convolutional networks you can definitely use a linear/cosine schedule for pretraining, and then choose the initial learning rate for linear-schedule-retraining adaptively, as in [1]. This will give much more realistic results.


### Minor Remarks
- Line 131: I presume it should be "**the** layer-wise pruning problem".
- In general, you do not seem to use the glossary package and define your DSF-acronym over and over again. That is a bit contrary to the purpose of an abbreviation. Also, you sometimes use DSF, and sometimes DFS (as in Double Factorization Sparse), see e.g. Line 315 or the caption in Line 686 where this happens in the same sentence.

#### References
[1] Zimmer, M., Spiegel, C., & Pokutta, S. (2021). How I Learned to Stop Worrying and Love Retraining. _arXiv preprint arXiv:2111.00843_. https://arxiv.org/abs/2111.00843

**Questions:**

- In Line 465 you state that your method does not support gradual pruning with fine-tuning between pruning steps, could you elaborate why? I am not sure what I am missing here.
- In Line 196, you first "look into the projection problem". I am not quite sure I understand correctly how that is not the entire problem? A proper solution to that is what you are looking for, isn't it?

---

> ### Author Response · Authors · 2024-11-15
> **Author's response**
>
> Thank you very much for a helpful and insightful review.
>
> Here are the responses to your concerns; we split them over multiple comments.
>
> >Section 3.1 would greatly benefit from a more detailed explanation of the ADMM method. How are Z and U initialized? I understand that it is not your job to explain ADMM in detail, but I think that nevertheless the paper would greatly benefit from more detailed remarks - at least in the appendix. Since this method is not standard (at least in the pruning literature and to my knowledge), I think it would be helpful to make this more clear.
>
> >In two sentences (Line 149, 150) you basically explain how you find the sparsity mask. Why do you precondition? How exactly is the cubic schedule (I presume Zhu & Gupta?) implemented, over how many iterations, with which interval between the increases? I am trying my best to infer this from somewhere, but it is nowhere to be found? Either I am missing something or the paper is lacking a crucial part, namely how the sparsity mask is found.
>
> We agree that the ADMM method for pruning is underexplored and not well known.
> We expanded section 3.1 to provide a quick overview of ADMM method used by "Fast and Effective Weight Update for Pruned Large Language Models" (https://openreview.net/forum?id=1hcpXd9Jir). Mainly, we talk about what ADMM does, how layer-wise pruning maps onto ADMM, and how the pruning mask is found.
> We decided not to put more details into an appendix because we believe that if the reader needs even more information, reading the original paper by Boza would be much more beneficial than reading the appendix in this paper.
>
> > In Line 258, you state that you are using the Wanda saliency map, I think it would be good to give the mathematical formulation to that, especially how you "scale one of the factors back".
>
> We added this to section 4.3.
>
> > Lines 37-39: If you replace the dense weight matrix with a product of two sparse matrices, will your model not be much slower at inference than when replacing with just a sparse matrix? For Low-rank decomposition, you at least get two linear layers which are much smaller dense matrices, but in your case, you basically have two sparse matrices. While you argue in Line 162 that the total number of multiplications is equal, this is far from realizable on the existing hardware. In practice, you incur a non-trivial overhead. I would like to hear the authors' thoughts on this.
>
> We added section 4.4 to discuss the DSF method's computational concerns. Part of our argument regarding speed can be summarized as follows:
>
> a) In many cases (e.g., local LLM inference), the main concern is fitting the best model into a small memory footprint. In this case, a slight decrease in inference speed might be tolerable.
>
> b) There are cases (again, for example, local single batch LLM inference) where the main computational bottleneck is transferring weights from memory to the local cache. This is explored in Flash-LLM (https://arxiv.org/pdf/2309.10285). Memory transfer is similar for DSF and for single sparsity.
>
> c) Doing two sparse multiplications is not much slower than doing one sparse multiplication with the same total number of nonzeros. We tested this using DeepSparse and found that DSF is ~10-20% slower than ordinary sparse multiplication (this is inline with other literature), but still faster than dense multiplication.
>
> > Line 50: "our method is the first layer-wise pruning method in which the larger pruned model is better than the dense smaller model" - Are you sure this is true? I feel like already the original SparseGPT paper gets fairly close and there have been a variety of improvements since then, e.g. using non-uniform layer-wise sparsity. Maybe this claim can or should be made more precise.
>
> SparseGPT was not better in terms of perplexity (5.63 for pruned Llama2-13B, 5.12 for dense Llama2-7B), but is better on zero-shot benchmarks. Outlier weighted sparsity (https://arxiv.org/pdf/2310.05175) might be better, but they do not report such results.
> "Discovering Sparsity Allocation for Layer-wise Pruning of Large Language Models" (https://openreview.net/forum?id=rgtrYVC9n4) reports better results in zero-shot benchmarks but does not report perplexity.
>
> Thus, we are adjusting our claim to uniform layer-wise pruning and perplexity measure.
>
> > Missing ablations: The paper is fixing a lot of hyperparameters and making claims without ablations. That includes e.g. the selection of sparsity distribution between the matrices (Line 209) or the initialization for A and B (Lines 248-250), among others. Such ablations should be added to justify the choice of parameters.
>
> We added even more ablations to the appendix.

---

> ### Author Response · Authors · 2024-11-15
> **Author response part 2**
>
> > Table 1: Why are you not comparing to SparseGPT, am I missing something? In my experience, SparseGPT is a very strong baseline. Also, why do you omit Wanda for 30% density? Is Wanda using a "finalization" step as well, i.e., are you reconstructing the remaining weights after pruning? You get that more or less for free if you pass the calibration data through anyway.
>
> ADMM by Boza is better than SparseGPT, so we feel that including it in Table 1 does not bring any value.
> We added SparseGPT results to Table 1. We omitted 30% density with Wanda, because it has bad results. But since this raised questions, we put it back. Wanda algorithm does not use finalization, it just selects weight to prune. But Wanda with finalization is just ADMM-1 algorithm in ADMM pruning paper and is generally worse than ADMM with pruning done gradually.
>
> > Section 5.4: I find the choice of hyperparameters for the retraining/fine-tuning quite arbitrary. You use a stepped schedule for most of the pretraining, then use a stepped learning rate schedule for retraining as wellf 70 total) epochs. [1] shows that if you properly choose the initial learning rate of a linear schedule, you can recover the accuracy drop of magnitude pruning in very few iterations. I am not sure if these results would withstand scrutiny. It would be good to use best practices here, i.e., for the convolutional networks you can definitely use a linear/cosine schedule for pretraining, and then choose the initial learning rate for linear-schedule-retraining adaptively, as in [1]. This will give much more realistic results.
>
> First of all, in the Imagenet experiment, we already used a linear schedule for fine-tuning (we also compared it to cyclical pruning to ensure that our baseline is high-quality).
>
> But you are completely right about CIFAR.
> Here, we switched to the linear learning rate schedule for finetuning, which leads to ~0.5% gains in almost all cases (and DSF is still better). Thank you for pointing this out. We keep the pretraining schedule as is since it produces high-quality dense results (slightly better than reported in the original resnet paper).
>
> >Line 131: I presume it should be "the layer-wise pruning problem".
> >In general, you do not seem to use the glossary package and define your DSF-acronym over and over again. ...
>
> Thank you for pointing all of this out. DFS is just a typo. We also removed unnecessary DSF definitions.
>
> > In Line 465 you state that your method does not support gradual pruning with fine-tuning between pruning steps, could you elaborate why? I am not sure what I am missing here.
>
> First of all, we changed the wording from "does not support" to "it is unclear how to do." Here is our reasoning why it is unclear:
> Imagine that your target density is 25%. In regular pruning, you can first prune to 50% density, then finetune network, and then prune again to 25% density. This works because you can easily prune the already-pruned matrix. But if you apply DSF, how should you apply DSF again with a lower target density? You could multiply the factors to get a dense matrix and factorize that, but is it the correct way?
>
>
> > In Line 196, you first "look into the projection problem". I am not quite sure I understand correctly how that is not the entire problem? A proper solution to that is what you are looking for, isn't it?
>
> The original layer-wise pruning problem is min ||XW - XAB||^2. The projection problem is min ||X - AB||^2, thus a simplified version.
>
>
> Thank you again for your very good suggestions. We hope our response clarifies your concerns. If that’s the case, we would greatly appreciate it if you would consider raising your score.

---

> > ### Comment · Reviewer_aP2Z · 2024-11-22
> >
> > Thank you for your response. I appreciate the revision of the paper, which makes a lot of things more accessible.
> >
> > > We added this to section 4.3.
> >
> > So are you here using the Hadamard product between a real value (the norm) and a matrix, or am I reading this in a wrong way?
> >
> > > We added section 4.4 to discuss the DSF method's computational concerns.
> >
> > I greatly appreciate the revisions, they should have been included in the original manuscript, since this setting is not as common. In my personal experience, executing two layers in a row (even with much smaller and dense matrices), leads to significant slowdowns. Also, up to 20% slowdown in your measurements honestly seems like a lot, given the perplexity improvement is not that dramatic. But I agree that this is up to the choice of the practitioner. I am also not entirely sure that the storage requirements are actually the same, given that starting from an $n \times m$ matrix, you end up with an $n \times n$ and an $n \times m$ matrix, i.e., you increase from $nm$ to $n^2+nm$ parameters, despite enforcing the same overall sparsity. I might be mistaken here, but e.g. for CSR format, the row pointer can also depend on the number of rows, which would clearly yield some overhead here.
> >
> > > ADMM by Boza is better than SparseGPT, so we feel that including it in Table 1 does not bring any value. We added SparseGPT results to Table 1. We omitted 30% density with Wanda, because it has bad results. But since this raised questions, we put it back. Wanda algorithm does not use finalization, it just selects weight to prune. But Wanda with finalization is just ADMM-1 algorithm in ADMM pruning paper and is generally worse than ADMM with pruning done gradually.
> >
> > Thank you for adding SparseGPT. I do not think that referring to another paper (which you apparently base on) and stating that ADMM is better, not requiring you to compare to SparseGPT, is really an option. The same holds for Wanda, how is the reader supposed to know that Wanda is ADMM-1 from the ADMM paper? Especially since you state that e.g. SparseGPT takes roughly half the time of your algorithm, and that time could be used to reconstruct the weights given the found sparsity mask of SparseGPT. In my experience, this improves even SparseGPT and would yield a more realistic comparison. It would be good to compare the methods on equal terms then. Anyways, thanks for clarification.
> >
> > > But you are completely right about CIFAR. Here, we switched to the linear learning rate schedule for finetuning, which leads to ~0.5% gains in almost all cases (and DSF is still better). Thank you for pointing this out.
> >
> > That is more clear now, thanks.
> >
> > > First of all, we changed the wording from "does not support" to "it is unclear how to do." Here is our reasoning why it is unclear: Imagine that your target density is 25%. In regular pruning, you can first prune to 50% density, then finetune network, and then prune again to 25% density. This works because you can easily prune the already-pruned matrix. But if you apply DSF, how should you apply DSF again with a lower target density? You could multiply the factors to get a dense matrix and factorize that, but is it the correct way?
> >
> > I see.
> >
> > > The original layer-wise pruning problem is min ||XW - XAB||^2. The projection problem is min ||X - AB||^2, thus a simplified version.
> >
> > Maybe I am missing something again, but I assume that then the projection problem should involve W and not X? But thanks for clarifying, I was not entirely sure whether I understood that distinction correctly.
> >
> > I thank the authors for their detailed answer and especially for revising the PDF accordingly. A major concern was the presentation of the work and I think that this has been properly addressed, as now many open gaps have been filled. Still, I am not entirely convinced that the setting of having two sparse matrices instead of a single one is interesting and relevant, especially given the fact that we are dealing with unstructured sparsity here and a) the improvements over existing algorithms are somewhat marginal and b) the costs of having two sparse matrices increase over having a single one. I will consider changing my score after the discussion phase.

---

> ### Author Response · Authors · 2024-11-22
> **Quick replies**
>
> Thank you for the replies, here is a quick clarification.
>
> > So are you here using the Hadamard product between a real value (the norm) and a matrix, or am I reading this in a wrong way?
>
> Yes, this was a slight abuse of notation, we changed to explicit element calculation in the paper. We calculate the norm of each feature and multiply the corresponding row in the matrix W (assuming your linear layer calculates $XW$, where $X$ is input and $W$ is the weight matrix).
>
> > I am also not entirely sure that the storage requirements are actually the same, given that starting from an matrix, you end up with an and an matrix, i.e., you increase from to parameters, despite enforcing the same overall sparsity. I
>
> We have the same number of nonzeros, so same storage requirement for them. Storage requirements for sparsity masks will be higher, but masks are a smaller part of the storage costs. We also have Figure 7 in the Appendix, which shows that DSF models are much better even when we consider overall storage size (not just the number of nonzeros).
>
> > I might be mistaken here, but e.g. for CSR format, the row pointer can also depend on the number of rows, which would clearly yield some overhead here.
>
> Yes, there will be a small overhead. For example, if you have a matrix size 4096*4096 and 95% sparsity, you need 838860 numbers for nonzeros and 838860 numbers for column indices and 4096 numbers for row pointers. DSF would need an additional 4096 number for row pointers in the second matrix, which is less than 0.3% of overall storage costs.
>
> > that time could be used to reconstruct the weights given the found sparsity mask of SparseGPT. In my experience, this improves even SparseGPT and would yield a more realistic comparison.
>
> SparseGPT does not have an explicit parameter that trades solution quality and solution time. There is a block size (the lower the blocksize, higher the solution time), but that does not lead to an increase in solution quality in our experience (and for example for Llama-2-7B and 50% sparsity changing block size from 128 to 64 lead to much worse solution, perplexity went from 6.52 to 6.99).
>
> > Maybe I am missing something again, but I assume that then the projection problem should involve W and not X?
>
> Sorry, this is our spelling error in that comment.
> It should read: "The projection problem is min ||**W** - AB||^2, thus a simplified version."
> Paper has this everywhere correctly, this was a mistake only in this comment.

---

> > ### Comment · Reviewer_aP2Z · 2024-11-22
> >
> > Thanks again for the answer. I am far from convinced that this reparametrization as a product of two sparse matrices will be relevant in the future, but since the authors put a lot of effort into improving the presentation as well as explaining the efficiency and storage issues, I will increase my score to 5, borderline reject.
> >
> > In case of acceptance, I highly recommend to further improve the readibility of the paper and to not leave out any discussion regarding the practical applicability of this setting, independent of how you solve it.

---

### Official Review · Reviewer_qugK · 2024-11-02

**Soundness:** 3
**Presentation:** 3
**Contribution:** 2
**Rating:** 8
**Confidence:** 4

**Summary:**

This work introduces double sparse factorization (DSF) which combines matrix decomposition with pruning to yield compressed neural networks with better generalization performance than the baselines used for comparison. The authors demonstrate that their proposed algorithm, an extension of the alternating direction method of multipliers (ADMM) algorithm, is capable of achieving competitive results when compressing pretrained LLMs and CNNs.

**Strengths:**

* While pruning of factors obtained from matrix decomposition is not a novel contribution per se (Le Magoarou & Gribonval, 2016), its application to pretrained model compression is novel as far as I know. In any case, this work clearly distinguishes itself from prior art by focusing on the model compression task, particularly in the context of LLMs.
* The paper is well written.
* The empirical results outperform strong, SOTA baselines in a variety of contexts for LLMs and CNNS.
* The authors take care to consider some of the practical concerns of their method, such as masking overhead. The demonstration of the generalization of DSF with a shared fixed A factor mask is particularly compelling.
* Compression and efficient inference is of particular importance as model sizes continue to grow and scale. As such, this work addresses a timely and important topic.

**Weaknesses:**

Overall, I am leaning towards accept. However, I have some significant concerns regarding the practical applicability of the proposed method. Fundamentally, we require compressed models that offer advantages in one or more of the following dimensions: memory overhead, latency, and/or throughput. For each of these dimensions, we can consider both training and inference. For the following discussion, let’s consider an intermediate fully-connected layer from a decoder block in a  LLaMa 2-7B @ 50% sparsity. This layer’s weight tensor is of shape (11008, 4096).

* Fine-tuning (FT) / training memory overhead: During FT, the proposed method requires ~37% more memory to store the intermediate activations of X@A@B compared with X@W. Activations can account for a significant portion of the overall memory footprint during training and this should be acknowledged in the paper.
* Mask overheads: Assuming a bit-mask compression strategy and no shared masks between A factors, we find a similar 37% increased overhead compared to single layer sparsity. With shared A factors this overhead drops to ~1%, assuming the mask is shared across all 36 decoder blocks. From this perspective, I find the fixed-mask variant of DSF to be the most practically interesting.
* Indices instead of bitmasks: In the introduction, the authors suggest using indices to store the locations of non-zero elements. However, given that we require uint16 indices to represent all positions in this weight tensor, this would only be practical at sparsities >= 15/16 compared to bit-masking. Given that this is currently an unobtainable level of sparsity for LLMs and roughly the limit at which we are able to find performant CNNs I find the suggestion to use indices to store non-zero locations poorly motivated.
* Latency and Throughput: This is the most challenging dimension to estimate. Although the FLOPs analysis suggests similar performance to OBC, this may be misleading considering the additional matmul operations required in the low-rank decomposition and subsequent increase in overall memory bandwidth required to store and load intermediate activations between subsequent matmul kernel calls. I would be more convinced of the practical application for DSF if the authors include a discussion on runtime latency. This could be supported by preliminary benchmarking using Neural Magic’s DeepSparse Engine which would offer some empirical evidence of improved runtime properties.
* 2:4 support: It’s unclear if the proposed method can support 2:4 sparsity as this would require a fixed sparsity level of 50% for both factors. The authors found that a smaller level of sparsity (~33%) yields the best performance but this prohibits using 50% sparsity in both factors as required for 2:4.
* Hyperparameter sensitivity: There are a number of specific sparsity values used in the experimental method (16% sparsity, 25% sparsity, etc.). How sensitive is DSF to these values? If DSF is applied to a new model family, is it required to perform a hyperparameter search to find the optimal sparsity level for the smaller factor? How were these sparsity levels found? Could the authors add the results of their hyperparameter sweep for these values, assuming this was how the values were determined.
* Reliance on PPL: The authors claim that their method “is the first layer-wise pruning method in which the larger pruned model is better than the dense smaller model.”. I believe this claim requires more evidence to support, namely, downstream evaluation for the compressed LLMs on real-world tasks. I would be more willing to support this claim with empirical results from the pruned models on OpenLLM Leaderboard v1 or similar. Relying on perplexity alone has been shown to be misleading for compressed models [1].
* LLM fine-tuning: The fine-tuning results section would benefit from expanding its scope to include fine-tuning of the compressed LLMs. I would also be interested to see what the memory overhead looks like for DSF when naive masked sparsity is used.

Based on the above it appears that the memory overhead with a fixed A factor mask is comparable to regular pruning. However, it appears likely that the latency for DSF will be worse than models pruned with other techniques (Wanda, etc.). It is unclear whether DSF can support 2:4 sparsity whereas other methods such as Wanda do support this format (albeit with high loss). If tuning is required per model to set the smaller factor sparsity, this may result in DSF being much more expensive to use on new models. I am willing to accept this paper as the generalization results are good and motivate future work exploring this direction. However, given that DSF is fundamentally motivated by network compression, a more holistic discussion of the above points would convince me to raise my score and, in my opinion, raise the impact of this work.

[1] A. Jaiswal, Z. Gan, X. Du, B. Zhang, Z. Wang, and Y. Yang, “Compressing LLMs: The Truth is Rarely Pure and Never Simple,” Oct. 02, 2023, arXiv: arXiv:2310.01382. doi: 10.48550/arXiv.2310.01382.

**Questions:**

* Specifically which LLaMa model is used for reporting the results in Table 1? The authors refer to both LLaMa 1 and 2 in their experimental setup.
* Does DSF provide latency/throughput benefits over dense or typical sparse networks (single layer sparsity) when using DeepSparse Engine?
* Can DSF be extended to 2:4 sparsity? What is the trade-off with generalization performance?
* How do the pruned LLMs compare when evaluated on OpenLLM v1 leaderboard evaluation tasks?
* Missing results for Wanda at 70% sparsity: Why were these not included in Table 1?
* What is the memory overhead when fine-tuning DSF LLMs in a naive way (i.e., with masked paramters intead of compressed representations)?

---

> ### Author Response · Authors · 2024-11-16
>
> Thank you very much for a helpful and insightful review.
>
> Here are the responses to your concerns; we split them over multiple comments.
>
> > I have some significant concerns regarding the practical applicability of the proposed method.
>
> We added section 4.4 to discuss DSF's computational concerns. We mainly argue that it is comparable to regular pruning, and it also gets much better results.
> Main arguments are summarized below as responses to your detailed questions.
>
> > Fine-tuning (FT) / training memory overhead: During FT, the proposed method requires ~37% more memory to store the intermediate activations of X@A@B compared with X@W. Activations can account for a significant portion of the overall memory footprint during training and this should be acknowledged in the paper.
>
> This is a very good point and completely true. One can mitigate the impact of this by using gradient checkpointing (we almost always use it when finetuning any LLM). We add simple experiments to the appendix, showing that DSF can be finetuned with inputs of similar size as regular pruning.
>
> > Mask overheads: Assuming a bit-mask compression strategy and no shared masks between A factors, we find a similar 37% increased overhead compared to single layer sparsity. With shared A factors this overhead drops to ~1%, assuming the mask is shared across all 36 decoder blocks. From this perspective, I find the fixed-mask variant of DSF to be the most practically interesting.
>
> Yes, masking overhead is ~37%. However, we also need to store the actual weight so that the overall model size increase is not that dramatic (e.g., it increases from 7.3 to 7.7 GiB for 50% pruned Llama2-7B). We discuss this in section 4.4 and also add a graph that compares actual model size with model perplexity to the experiments section. It shows that DSF is much better than regular layer-wise pruning for any storage size.
>
> > Indices instead of bitmasks: In the introduction, the authors suggest using indices to store the locations of non-zero elements. However, given that we require uint16 indices to represent all positions in this weight tensor, this would only be practical at sparsities >= 15/16 compared to bit-masking. Given that this is currently an unobtainable level of sparsity for LLMs and roughly the limit at which we are able to find performant CNNs I find the suggestion to use indices to store non-zero locations poorly motivated.
>
> Yes, you are completely right. We removed this passage from the introduction.
>
> > Latency and Throughput: This is the most challenging dimension to estimate. Although the FLOPs analysis suggests similar performance to OBC, this may be misleading considering the additional matmul operations required in the low-rank decomposition and subsequent increase in overall memory bandwidth required to store and load intermediate activations between subsequent matmul kernel calls. I would be more convinced of the practical application for DSF if the authors include a discussion on runtime latency. This could be supported by preliminary benchmarking using Neural Magic’s DeepSparse Engine which would offer some empirical evidence of improved runtime properties.
>
> > Does DSF provide latency/throughput benefits over dense or typical sparse networks (single layer sparsity) when using DeepSparse Engine?
>
> Yes, the runtime is a main concert with any sparsified neural networks. We added a discussion about this to the paper. Main summary:
>
> a) In many cases (e.g., local LLM inference), the main concern is fitting the best model into a small memory footprint. In this case, a slight decrease in inference speed might be tolerable.
>
> b) There are cases (again, for example, local single batch LLM inference) where the main computational bottleneck is transferring weights from memory to the local cache. This is explored in Flash-LLM (https://arxiv.org/pdf/2309.10285). Memory transfer is similar for DSF and single sparsity.
>
> c) Doing two sparse multiplications is not much slower than doing one sparse multiplication with the same total number of nonzeros. We tested this using DeepSparse and found that DSF is ~10-20% slower than ordinary sparse multiplication (this is inline with other literature), but still faster than dense multiplication.

---

> ### Author Response · Authors · 2024-11-16
> **Authors' response part 2**
>
> > 2:4 support: It’s unclear if the proposed method can support 2:4 sparsity as this would require a fixed sparsity level of 50% for both factors. The authors found that a smaller level of sparsity (~33%) yields the best performance but this prohibits using 50% sparsity in both factors as required for 2:4.
>
> > Can DSF be extended to 2:4 sparsity? What is the trade-off with generalization performance?
>
> DSF cannot do 2:4 sparsity right now. Also, if both factors are in 2:4 sparsity and DSF runs over a square matrix, then you have the same number of nonzeros as the original matrix, which does not bring any gains. What might be viable with 2:4 are two things:
>
> a) Having one factor in 2:4 format and the second one as very sparse (95+% of sparsity).
>
> b) Having block sparsity with blocks having 2:4 sparsity.
>
> But both options require significant research and tuning, and we leave them for future work right now.
>
> > Hyperparameter sensitivity: There are a number of specific sparsity values used in the experimental method (16% sparsity, 25% sparsity, etc.). How sensitive is DSF to these values? If DSF is applied to a new model family, is it required to perform a hyperparameter search to find the optimal sparsity level for the smaller factor? How were these sparsity levels found? Could the authors add the results of their hyperparameter sweep for these values, assuming this was how the values were determined.
>
> We found that having both factors with an equal number of nonzeros works quite well but can be slightly tuned. We added more ablations to the appendix (we determined hyperparameters by taking a couple of layers and observing reconstruction error).
>
> > Reliance on PPL: The authors claim that their method “is the first layer-wise pruning method in which the larger pruned model is better than the dense smaller model.”. I believe this claim requires more evidence to support, namely, downstream evaluation for the compressed LLMs on real-world tasks. I would be more willing to support this claim with empirical results from the pruned models on OpenLLM Leaderboard v1 or similar. Relying on perplexity alone has been shown to be misleading for compressed models [1].
>
> > How do the pruned LLMs compare when evaluated on OpenLLM v1 leaderboard evaluation tasks?
>
> We are adjusting our claim to uniform layer-wise pruning and perplexity measure.
> We also added measurements on seven zero-shot evaluations used in other pruning papers before (arc-easy, arc-challenge, winogrande, and hellaswag are also parts of OpenLLM v1).
>
> > LLM fine-tuning: The fine-tuning results section would benefit from expanding its scope to include fine-tuning of the compressed LLMs.
>
> A decent full-parameter finetuning run takes a lot of time (one also needs to do a decent hyperparameter sweep, at least for the learning rate). We are unsure whether we can make this happen during the discussion period.
>
> > I would also be interested to see what the memory overhead looks like for DSF when naive masked sparsity is used.
>
> Using masked sparsity naively would result in many problems (mainly ~40% more space needed for weight storage). We tested a different approach, where we stored weight in compressed format and unpacked on the fly during the forward pass. If finetunng batch is large enough, this occurs a negligible time overhead. We explore this more in the appendix A.4.
>
> > Specifically which LLaMa model is used for reporting the results in Table 1? The authors refer to both LLaMa 1 and 2 in their experimental setup.
>
> We have Llama1-7B (denoted as 1-7B) and Llama2-7/13/70B (denoted as 2-7B, 2-13B, 2-70B respectively).
>
> > Missing results for Wanda at 70% sparsity: Why were these not included in Table 1?
>
> We omitted 30% density with Wanda because it had bad results. But since this raised questions, we put it back.
>
> Thank you again for your very good suggestions. We hope our response clarifies your concerns. If that’s the case, we would greatly appreciate it if you would consider raising your score.

---

> ### Author Response · Authors · 2024-11-25
> **Additional results for LLM finetuning (distillation)**
>
> We finished limited finetuning (distillation) experiments (run for 2 days on 4xA100).
> We distilled Llama2-7B and compared the fine-tuning of regularly pruned model via ADMM with 50% density and the DSF pruned model with 45% density (after accounting for masks, these models have similar storage sizes).
> Results are summarized in the appendix, and here we put the table for the reference:
>
> | Model     | Pruning type | PPL w/o finetuning | Zero-shot w/o finetune | PPL w/ finetuning | Zero-shot w/ finetuning |
> |-----------|--------------|--------------------|------------------------|-------------------|-------------------------|
> | Llama2-7B | Dense        | 5.12               | 59.71                  |  -                 |   -                      |
> | Llama2-7B | ADMM 50%     | 6.33               | 56.64                  | 5.61              | 58.00                   |
> | Llama2-7B | DSF 45%      | 5.78               | 57.03                  | 5.35              | 59.00                   |
>
> As we can see, DSF retains its advantage even after finetuning.

---

> > ### Comment · Reviewer_qugK · 2024-11-27
> > **Deepsparse benchmark clarifications**
> >
> > Thanks for the detailed rebuttal.
> >
> > > c) Doing two sparse multiplications is not much slower than doing one sparse multiplication with the same total number of nonzeros. We tested this using DeepSparse and found that DSF is ~10-20% slower than ordinary sparse multiplication (this is inline with other literature), but still faster than dense multiplication.
> >
> > In Table 6 I see the comparison with the single-sparse matrix. Could you share the dense runtimes as well (ideally wrapped with deep sparse engine or torch.compile(mode='max-autotune')? Some of the features of deepsparse engine will also benefit the dense model so it's important to compare these settings explicitly.
> >
> > From Table 6 caption:
> > > Each time, we compare the runtime of 48 layers with simple sparsity or DSF with an equal number of nonzeros (i.e., running 96 layers with half the density)
> >
> > This setting is somewhat different than what DSF proposes. Could you share deepsparse benchmarks with alternating 16/25% sparsity and the remainder non-zeros in even-numbered factors allocated based on global sparsity target as discussed in this paragraph:
> >
> > > When factorizing square matrices (mainly in self-attention), we set the sparsity of one sparse factor to 16%. When factorizing rectangular matrices, the smaller factor will have 25% sparsity. The number of nonzeros in the other factor is just the target number of nonzeros minus the number of nonzeros in the first factor.
> >
> > Could you repeat the deepsparse benchmarks with additional cores as well? It would be beneficial to see the multi-threaded runtimes too.
> >
> > The downstream evals are great to see and much appreciated as is the formal discussion of computational considerations in the main body of the paper.
> >
> > I remain concerned that this work has limited practical relevance due the increased mask overhead and latency compared to single-sparse networks. If the authors can response to the above requests and if the comparisions with dense appear to show at least modest latency benefits I would be willing to improve my score.

---

> ### Author Response · Authors · 2024-11-28
> **Clarifications**
>
> Thank you, for the response, here is a quick clarification.
>
> >Could you share the dense runtimes as well.
>
> They were actually in Table 6, but hidden in parentheses under batch size (not the best idea on our part). We put dense runtimes into a separate column to make benchmarks more readable. We also double-checked the dense runtime and found that Deepsparse is slightly faster than just torch.compile with max-autotune (and thus every dense number is from Deepsparse).
>
> In all cases, even at 50% density, DSF is faster than dense runtime (e.g., for 4096x4096 matrices with 64 batch size, we have dense runtime of 872 ms, simple sparsity runtime of 470 ms, and DSF runtime of 533 ms).
>
> > Could you repeat the deepsparse benchmarks with additional cores as well?
>
> Added Table 7 with this.
>
> > Could you share deepsparse benchmarks with alternating 16/25% sparsity...
>
> We tested this and it is not different from the results with even distribution.
> For square matrices with size 4096x4096 with 50% total density, 16% density in the first factor and 34% in the second we get following:
>
> | Batch size | Dense runtime | Single sparsity | DSF |
> |------------|---------------|-----------------|-----|
> | 64         | 162           | 101             | 122 |
> | 256        | 551          | 301             | 354 |
>
>
>
>
>
> For rectangular matrices with size 4096*11008 (found in Llama-2-7B MLP block), 50% total density, 25% density in smaller factor and 40.7% density in the larger factor we get following runtimes:
>
> | Batch size | Dense runtime | Single sparsity | DSF |
> |------------|---------------|-----------------|-----|
> | 64         | 426           | 251             | 276 |
> | 256        | 1474          | 792             | 929 |
>
> >  limited practical relevance
>
> We found that in almost all scenarios we tested so far (even outside ones mentioned in the paper), DSF is better than regular sparsity and can have surprisingly good results.
>
> Let us share a **very preliminary** test we ran a couple of days ago (we are definitely not putting this into current paper). We aggressively pruned Llama3-8B to ~16% density, with the goal that the final result would have 2bits per parameter (8bits would be used for nonzeros and mask will be compressed with something similar to compression in https://proceedings.mlsys.org/paper_files/paper/2024/file/c74b624843218d9b6713fcf299d6d5e4-Paper-Conference.pdf) .
> We then fine-tuned models for half of a day and measured perplexity.
>
> When we compare with results from PV-tuning paper (https://arxiv.org/abs/2405.14852 table 2), we would find following
>
> | Method | Perplexity |
> |-----|-----|
> | Dense | 5.54 |
> | QUIP | 76.95 |
> | Regular sparsity | 16.3 |
> | DB-LLM | 12.77 |
> | DSF | 10.3 |
> | PV-tuning | 6.99 |
>
> As you can see, DSF can be almost as good as the best quantization method (and is better than many good quantization methods that use fine-tuning like DB-LLM) while offering the benefits of relatively easy fine-tuning. Also keep in mind, this was just a first very quick experiment with setup like this one.

---

> > ### Comment · Reviewer_qugK · 2024-11-29
> >
> > Thanks for these further clarifications. In my opinion, you have presented sufficient preliminary evidence that this approach could lead to practical benefits. As such, I have increased my score to 8.

---

> ### Author Response · Authors · 2024-11-29
>
> Thank you very much for raising the score and for the very good feedback.

---

### Official Review · Reviewer_VFC3 · 2024-11-04

**Soundness:** 3
**Presentation:** 3
**Contribution:** 3
**Rating:** 6
**Confidence:** 4

**Summary:**

The paper proposes Double Sparse Factorization (DSF) of the weight matrices to prune them effectively. They formulate it as an alternating optimization and optimize using ADMM. The experiments show a clear benefit of the proposed method on Llama for a language task and resnet on image classification.

**Strengths:**

1. The idea is nice, the problem formulation is neat, and using ADMM for optimization is elegant.
2. On pruning LLAMA, the method shows clear benefit over the compared methods. Image classification experiments are marginally better than previous methods.

**Weaknesses:**

1. The SVD comparison is unfair in my opinion. SVD is more suited for low-rank compression and it may not enforce sparsity. Using the sparsity ratio as the main criterion may not be ideal. Why not use FLOPs? As FLOPS directly relates to inference speed as opposed to sparsity ratio. I would suggest that the authors include a comparison based on FLOPs in addition to the sparsity ratio. This would provide a more comprehensive evaluation of computational efficiency across different compression methods, including SVD and sparse factorization approaches.
2. ADMM optimization may be compute-intensive. Not much discussion about it unless I missed something. Could you provide an asymptotic time complexity analysis and/or empirical running time comparison of the ADMM? You may also discuss the trade-offs between computational cost and compression quality, as it would give readers a clearer understanding of practical applicability of the proposed method.

**Questions:**

1. Could you discuss how this is related to sparse coding?

---

> ### Author Response · Authors · 2024-11-16
> **Authors' response**
>
> Thank you for your thoughful review.
>
> Here are the responses to your concerns:
>
> >The SVD comparison is unfair in my opinion. SVD is more suited for low-rank compression and it may not enforce sparsity. Using the sparsity ratio as the main criterion may not be ideal. Why not use FLOPs? As FLOPS directly relates to inference speed as opposed to sparsity ratio. I would suggest that the authors include a comparison based on FLOPs in addition to the sparsity ratio. This would provide a more comprehensive evaluation of computational efficiency across different compression methods, including SVD and sparse factorization approaches.
>
> Yes, SVD comparison is kind of unfair; that's why we compare with it only in the section about the comparison of matrix approximation methods.
>
> We are not using FLOPs since FLOPs are in a linear relationship with sparsity in the case of single-layer processing and also when using uniform sparsity over all layers (as is in the case of LLMs).
> FLOPs do not have a direct relationship with sparsity in the case of nonuniform sparsity in vision models (different layers process different numbers of elements). That's why we use FLOPs in the OBC section.
>
> SVD has the obvious benefit of doing just two dense matrix multiplications and thus not having time overhead associated with sparse matrix multiplication. But, our primary baseline is regular pruning, which already has sparse matrix multiplication. We added section 4.4 to discuss the computational concern of DSF and argue that DSF has similar overheads as regular pruning.
>
> > ADMM optimization may be compute-intensive. Not much discussion about it unless I missed something. Could you provide an asymptotic time complexity analysis and/or empirical running time comparison of the ADMM? You may also discuss the trade-offs between computational cost and compression quality, as it would give readers a clearer understanding of practical applicability of the proposed method.
>
> The original ADMM paper (https://openreview.net/forum?id=1hcpXd9Jir) shows that ADMM is better than solving this problem via gradient descent and is definitely better than $n$ independent linear regressions. It also shows that it is as fast as SparseGPT.
>
> At the end of section 5.1, we mention the pruning time. We added one more experiment to the appendix, showing the relationship between pruning time and model quality.
>
> > Could you discuss how this is related to sparse coding?
>
> We added this to the related work section.
>
> We hope that these answers clarify your concerns.

---

> ### Author Response · Authors · 2024-11-29
> **One more note**
>
> One more note:
>
> We have moved the whole section about layer-wise error comparison (comparing DSF with magnitude pruning, SVD, ...) to the appendix.
>
>
> Is there anything more we can do to clarify your concerns?

---

### Author Response · Authors · 2024-12-02
**Global response and summary**

We sincerely thank all reviewers for their constructive and thoughtful feedback.
Reviewers agreed that our Double Sparse Factorization (DSF) produces strong SOTA results for model pruning.
They also praised the novelty of the idea, paper clarity, and ADMM usage.

The reviewers' main concern was the computation overhead of the DSF. We added whole section 4.4 to discuss this in deeper detail and summarize the main points as follows:

* **Memory storage overhead.** Storing two masks requires more space than storing one mask (2x more for square matrices, less for rectangular ones). However, for small-medium sparsities (50-75%), this cost is significantly smaller than the cost of storing nonzero values.
We also added Figure 3 to compare the total model storage size and perplexity for various sparsities. DSF is still better than the best layer-wise pruning method.

* **Compute time overhead.** Doing two sparse multiplications instead of one (with the same total number of nonzeros) incurs non-trivial overhead. We measured this on the CPU and saw a 10-20% increase in running time. However, on the DeepSparse engine, even DSF with 50% of nonzeros was still faster than a dense model.

---

### Meta-Review · Area_Chair_DNQ8 · 2024-12-20

**Metareview:**

The authors propose a method for sparsifying neural network parameter matrices by reparameterizing them as the product of two sparse matrices.  This is accomplished via a heuristic that seeks to minimize the error of the factorized approximation relative to the original weights subject to hard (i.e., L0) sparsity constraints on the factorized matrices via ADMM.  While the idea is conceptually rather simple, the authors are largely in agreement that the empirical performance of the method is convincing.  Reviewers raised concerns that the increased overhead of performing two sparse matrix multiplications could be detrimental, but the authors note that their primary goal is to reduce the memory footprint of the model and not necessarily improve inference speed.

While one reviewer still notes potential issues with clarity in the manuscript, this would appear to be something that can be addressed in a final revision and I believe this work is of sufficient interest and quality to be accepted.

**Additional Comments On Reviewer Discussion:**

The authors were largely responsive to the initial reviews, which led to two reviewers raising their initial scores.  One reviewer still has concerns regarding the presentation of some topics of the work, which the authors should seek to address when preparing a final version of the manuscript.

---

### Decision · Program_Chairs · 2025-01-22

Accept (Poster)